# FLOPS: FORWARD LEARNING WITH OPTIMAL SAMPLING

**Tao Ren**[1,2*]**, Zishi Zhang**[1,2*]**, Jinyang Jiang**[1,2] **, Guanghao Li**[2] **, Zeliang Zhang**[3] **,**
**Mingqian Feng**[4] **, Yijie Peng**[1,2†]
[1] Guanghua School of Management, Peking University     [2] Xiangjiang Laboratory
[3] Tsinghua Shenzhen International Graduate School, Tsinghua University
[4] Huazhong University of Science and Technology     [5] Johns Hopkins University
{rtkenny,zishizhang,jinyang.jiang}@stu.pku.edu.cn
ligh24@mails.tsinghua.edu.cn, hust0426@gmail.com, mfeng10@jh.edu,
pengyijie@pku.edu.cn

## ABSTRACT

Given the limitations of backpropagation, perturbation-based gradient computation methods have recently gained focus for learning with only forward passes, also referred to as queries. Conventional forward learning consumes enormous queries on each data point for accurate gradient estimation through Monte Carlo sampling, which hinders the scalability of those algorithms. However, not all data points deserve equal queries for gradient estimation. In this paper, we study the problem of improving the forward learning efficiency from a novel perspective: how to reduce the gradient estimation variance with minimum cost? For this, we allocate the optimal number of queries within a set budget during training to balance estimation accuracy and computational efficiency. Specifically, with a simplified proxy objective and a reparameterization technique, we derive a novel plug-and-play query allocator with minimal parameters. Theoretical results are carried out to verify its optimality. We conduct extensive experiments for fine-tuning Vision Transformers on various datasets and further deploy the allocator to two black-box applications: prompt tuning and multimodal alignment for foundation models. All findings demonstrate that our proposed allocator significantly enhances the scalability of forward-learning algorithms, paving the way for real-world applications. The implementation is available at https://github.com/RTkenny/FLOPS-Forward-Learning-with-OPtimal-Sampling.

## 1 INTRODUCTION

Ever since the success of backpropagation (BP) (Rumelhart et al., 1986), researchers have sought alternate methods to bypass the iterative computation of the backward pass for the sake of efficiency, interpretability, and biological plausibility (Ma et al., 2020; Nøkland, 2016; Jacot et al., 2018). Moreover, existing practical scenarios urge for scalable forward learning methods, when the black-box nature renders taking the derivative infeasible or difficult. For example, as the model parameters increase, machine learning (ML) systems usually integrate with third-party black-box APIs (Achiam et al., 2023).

There have been continuous efforts to develop learning algorithms that rely solely on the forward pass (Lillicrap et al., 2020). The forward learning uses multiple queries on one data point to estimate the gradient (Spall, 1992; Zhang et al., 2024). The high dimensionality and the rugged loss landscape of ML problems pose arduous challenges for deriving a low-variance gradient estimator. Current literature has demonstrated that a sufficient number of queries on each data point is necessary for successful training (Chen et al., 2023; Zhang et al., 2024). To achieve good gradient estimation and task performance, all existing methods equally assign a large number of queries to different data points, neglecting the varying difficulty of gradient estimation at these data.

---

[*] These authors contributed equally to this work.
[†] Corresponding author.

However, it is not always helpful for the forward learning algorithms to increase the number of queries. Especially for the large model training on a large dataset, it usually requires an extensive number of queries, which poses great challenges to the memory and computation cost. For example, The forward-forward (FF) (Hinton, 2022) algorithm and the feedback alignment (FA) (Nøkland, 2016) are only capable of training multilayer perceptrons on MNIST (LeCun, 1998) or CIFAR (Krizhevsky et al., 2009). To scale up the forward algorithm to large model training on large datasets, DeepZero (Chen et al., 2023) and Mezo (Malladi et al., 2023) explore the use of simultaneous perturbation stochastic approximation (SPSA) (Spall, 1992) in training the convolution network from scratch and fine-tune the language model, respectively. But with such "large" scale optimization problem, DeepZero requires 192 queries to train a pruned ResNet-20 with only $12K$ parameters. Mezo needs meticulously selected prompts to ensure performance, and its gradient estimator is too noisy due to insufficient query numbers, thus suffering from highly unstable performance.

There is a great challenge required to be considered carefully in forward learning. On the one hand, consuming large amounts of queries on every data point results in high computational costs and long training time. On the other hand, using insufficient queries can save memory and computational costs but lead to unstable performance. Efficiently utilizing the queries with limited memory and computation budget is at the core of the scalability for forward learning to reduce the gradient estimation variance and improve the model training efficiency.

To achieve a good balance between the query cost and gradient estimation variance, we propose to study the query allocation problem under the forward learning framework. In our paper, we first unify the different estimators under the perturbation-based framework and formulate the allocation problems. Then, we propose the optimal allocation by constructing a query budget allocator to assign queries adaptively following a data-aware paradigm. An overview of our method and the comparison with previous work can be found in Figure 1. Specifically, our contribution can be summarized as follows,

1. We unify different BP-free methods under the framework of perturbation-based optimization. We are the first to formulate the query allocation problem for forward learning in the literature.

2. We find a simplified objective for the allocation problem and solve it via the likelihood ratio method in a lightweight style. With the optimal allocation, we can compress the queries to the minimum number of 20.

3. We theoretically deduce that the enhancement in allocation is assured by a defined lower bound. The allocator would assign the queries for different data in a mini-batch according to the variance of the gradient estimation. Intuitively, the estimator with high variance would be assigned with more queries, and vice versa.

4. We conduct extensive experiments to fine-tune Vision Transformer (Dosovitskiy, 2020) on challenging datasets, including ImageNet-1K Deng et al. (2009), achieving state-of-the-art performance in forward learning. Furthermore, we apply the proposed method to two real-world black-box problems: black-box prompt tuning and alignment between foundation models. Our approach demonstrates scalability and efficiency improvements over all baselines, including backpropagation (BP).

## 2 RELATED WORK

**Backpropagation-free learning.** People have explored various algorithms without backpropagating the error (Wierstra et al., 2014; Frenkel et al., 2021; Journé et al., 2022; Kao & Hariharan, 2024). Frenkel et al. (2021) eliminates the feedback pathway by leveraging fixed random projections of target labels to hidden layers. Kao & Hariharan (2024) employs a dual-network structure to propagate input and target signals in anti-parallel directions. Forward gradient (Baydin et al., 2022; Silver et al., 2021), applying the chain rule from a different direction than BP, utilizes forward-mode automatic differentiation (AD) to train the ML model. Forward-mode AD needs specific AD software and full access to the model structure. Therefore, training the deep model under black-box scenarios is not feasible by the forward gradient (Ren et al., 2022).

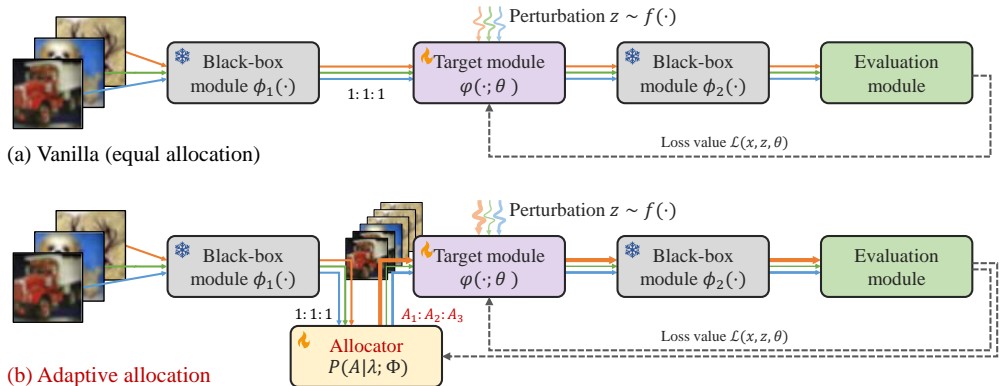

Figure 1: Illustration of allocating the query budget of the forward learning paradigm. As shown in (a), previous methods equally allocate the query across different data. Our method, as shown in (b), adaptively allocate the queries under theoretically guaranteed optimality.

**Perturbation-based forward learning.** Built on the theory of stochastic optimization, researchers have devised surrogate methods to approximate the first-order gradients, such as SPSA (Spall, 1992). The SPSA injects noise with opposite signs into the model parameters to form a finite-difference style estimator. Recently, Mezo (Malladi et al., 2023) applied SPSA to fine-tune the language model. DeepZero (Chen et al., 2023) integrates SPSA with sparsity and trains the network from scratch with many queries. However, their methods still have some limitations and their query budgets are equally allocated, neglecting the data difference. Jiang et al. (2023) extends the likelihood ratio (LR) estimator to train a wide range of neural networks, whereas LR has not been verified on modern-scale architectures, such as Transformers (Vaswani, 2017), leaving a significant gap for further research to improve the scalability of forward learning.

**Machine learning for black box scenarios.** The applications to real-world black-box scenarios call for scalable BP-free algorithms. When deploying ML algorithms on specific hardware systems, computation resources are limited and BP becomes infeasible. In the fields of physics and chemistry, ML models have to interact with environments that include non-differentiable operations (Momeni et al., 2023; Gu et al., 2021). Moreover, extremely large models are often only accessible through third-party API (Sun et al., 2022). Evolutionary algorithms and reinforcement learning have been applied to tune the prompt for the black-box language model (Diao et al., 2022).

**Advanced technique for data sampling and reshuffling.** In the literature on stochastic optimization, apart from uniform sampling, people have designed various methods to sample data points from the dataset by importance sampling (Zhao & Zhang, 2015), reshuffling (Lu et al., 2022), etc. Their method can reduce the variance of the stochastic gradient. Their methods share some similarities with our OPtimal Sampling. However, the key distinction between our work and the existing methods lies in the probability spaces considered. While the existing methods focus on reducing variance in the probability space associated with "sampling data points from the dataset", our approach focuses on perturbation-based forward learning, where the variance lies in the probability space of the injected noise for gradient estimation.

## 3 A UNIFIED PERSPECTIVE OF PERTURBATION-BASED FORWARD LEARNING

Given a generic neural network with no assumption on its architecture, the input is denoted as $x \in \mathbb{R}^{d_x}$, and the output is given by $y = \phi_2(\varphi(\phi_1(x); \theta)) \in \mathbb{R}^{d_y}$, where $\phi_1(\cdot)$ and $\phi_2(\cdot)$ are black-box parts, and $\varphi(\cdot; \theta)$ is the targeted module with trainable parameters $\theta \in \Theta \subset \mathbb{R}^{d_\theta}$. Conducting backpropagation through $\phi_1(\cdot)$ and $\phi_2(\cdot)$ is infeasible or difficult. Training the neural network is to solve such an optimization problem: $\min_{\theta \in \Theta} \frac{1}{|\mathcal{X}|} \sum_{x \in \mathcal{X}} \mathcal{L}(y)$, where $\mathcal{X}$ is the dataset for training, and $\mathcal{L}(\cdot)$ is an evaluation procedure yielding the loss feedback. The growing scale of parameters in modern neural networks and the dataset makes gradient-based methods the only feasible solution. Since forward learning do not rely on the chain rules to compute the true gradient, the primary challenge remains in estimating the gradient accurately and efficiently.

In addition to approaches requiring computation graphs like BP, forward-learning algorithms estimate gradients by perturbing the neural activity (intermediate output or parameter of the targeted module $\varphi(\cdot; \theta)$) and observing the impact of the injected perturbation on the final loss value, thereby solving the stochastic version of the problem: $\min_{\theta \in \Theta} \frac{1}{|\mathcal{X}|} \sum_{x \in \mathcal{X}} \mathbb{E}_{z \sim f(\cdot)}[\mathcal{L}(y)]$, where $z$ is the injected noise with a density $f(\cdot)$.

Forward learning algorithms can be divided into two categories: the LR family and the SPSA/ES family. The LR family perturb the intermediate output of the targeted module to perform estimation Jiang et al. (2023), i.e., $y = \phi_2(\varphi(\phi_1(x); \theta) + z)$. The gradient can be estimated by

$$\nabla_\theta \mathbb{E}[\mathcal{L}(y)] = \mathbb{E}_{z \sim f(\cdot)}[G_{LR}(x, \theta, z)] \triangleq \mathbb{E}_{z \sim f(\cdot)}[-\mathcal{L}(y) D_\theta^\top \varphi(\phi_1(x); \theta) \nabla_z \ln f(z)], \quad (1)$$

where $D_b a \in \mathbb{R}^{d_a \times d_b}$ denote the Jacobian matrix of $a \in \mathbb{R}^{d_a}$ with respect to $b \in \mathbb{R}^{d_b}$.

The SPSA/ES family injects noise to the parameters (Spall, 1992; Salimans et al., 2017), i.e., $y = \phi_2(\varphi(\phi_1(x); \theta + z))$. For continuous noise distributions, the gradient estimation can be derived as a special case of equality (1) as follows:

$$\nabla_\theta \mathbb{E}[\mathcal{L}(y)] = \mathbb{E}_{z \sim f(\cdot)}[G_{ES}(x, \theta, z)] \triangleq \mathbb{E}_{z \sim f(\cdot)}[-\mathcal{L}(y) \nabla_z \ln f(z)], \quad (2)$$

which takes the same form termed as the evolutionary strategies (ES) estimation (Salimans et al., 2017). The SPSA is a special case of ES which uses Gaussian noise $z \sim \mathcal{N}(0, \sigma^2 I_d)$ and applies an antithetic variable trick for variance reduction to equality (2). The SPSA can be interpreted as randomized finite differences in high-dimensional space. It takes the form

$$\begin{aligned}\nabla_\theta \mathbb{E}[\mathcal{L}(y)] &= \mathbb{E}_{z \sim \mathcal{N}(\cdot)}[G_{SPSA}(x, \theta, z)] \\ &= \mathbb{E}_{z \sim \mathcal{N}(\cdot)}\big[\frac{z}{2\sigma}(\mathcal{L}(\phi_2(\varphi(\phi_1(x); \theta + z))) - \mathcal{L}(\phi_2(\varphi(\phi_1(x); \theta - z))))\big].\end{aligned} \quad (3)$$

To estimate the expectation of gradient according to equation (1, 2 and 3), both SPSA/ES and LR family use Monte Carlo sampling. With $A > 0$ queries of Monte-Carlo sampling to estimate the gradient on data point $x$, the corresponding estimations take the sample-average form $\nabla_\theta \hat{\mathcal{L}}(y) = \frac{1}{A} \sum_{i=1}^A G(x, \theta, z_i)$, where $z_i$ is sampled independently from $f(\cdot)$ and $G(x, \theta, z_i)$ is the $i$th sample of the corresponding gradient estimator ($G(x, \theta, z_i)$ can be $G_{LR}(x, \theta, z_i)$, $G_{ES}(x, \theta, z_i)$ or $G_{SPSA}(x, \theta, z_i)$).

The parameter is updated iteratively with a mini-batch of data points randomly drawn from $\mathcal{X}$, denoted as $\{x_1, \cdots, x_B\}$, where $B$ is the batch size. We denote the gradient estimation on data $x_j$ as $\nabla_\theta \hat{\mathcal{L}}_j(\theta) = \frac{1}{A_j} \sum_{i=1}^{A_j} G(x_j, \theta, z_{i,j})$, where $A_j$ denotes the allocated query size, and the batch-wise gradient estimation is $\nabla_\theta \hat{\mathcal{L}}(\theta_t) = \frac{1}{B} \sum_{j=1}^B \nabla_\theta \hat{\mathcal{L}}_j(\theta_t)$. The updating recursion can be written as

$$\theta_{t+1} = \theta_t - \eta_t \nabla_\theta \hat{\mathcal{L}}(\theta_t) = \theta_t - \eta_t \frac{1}{B} \sum_{j=1}^B \nabla_\theta \hat{\mathcal{L}}_j(\theta_t) = \theta_t - \eta_t \frac{1}{B} \sum_{j=1}^B \frac{1}{A_j} \sum_{i=1}^{A_j} G(x_j, \theta, z_{i,j}), \quad (4)$$

where $\theta_t$ is the network parameter at the $t$-th step, and $\eta_t$ is the learning rate.

By default, the query amount $A_j$ allocated to estimate the gradient associated with each data point $x_j$ is equally allocated. However, the difficulty of gradient estimation and the model performance differ over data points. To conserve computational resources while maintaining the quality of gradient estimation, the number of queries should be properly allocated across data within the mini-batch. Utilizing the likelihood ratio technique, we propose a universal accelerator for all perturbation-based training methods from a data-aware perspective.

## 4 OPTIMAL SAMPLING VIA LIKELIHOOD RATIO METHOD

### 4.1 GAUSSIAN ALLOCATOR AND BERNOULLI ALLOCATER

**Definition.** The query allocator is defined as a random vector $\boldsymbol{A} = (A_1, \cdots, A_B)$, where each component $A_j$ represents the (relative) number of queries allocated to the $j$-th data point in the batch.

The allocator is assumed to follow a probability measure $\boldsymbol{A} \sim P(\boldsymbol{A}|\boldsymbol{\lambda}; \Phi)$, which is parameterized by $\boldsymbol{\lambda} \in \Lambda$ and conditioned on features $\Phi$, such as the loss. At each step of neural network training, a sample of $\boldsymbol{A}$ is drawn from this distribution, and queries are allocated to data points based on the sample. In this work, we select a Gaussian Allocator (GA)

$$\boldsymbol{A} \sim N(\boldsymbol{\mu}, \Sigma|\boldsymbol{\lambda}; \Phi), \tag{5}$$

where $\boldsymbol{\lambda} = (\boldsymbol{\mu}, \Sigma) \in \mathbb{R}^B \times \mathbb{R}^{B \times B}$. The matrix $\Sigma$ captures the correlation structure between data points and facilitates the allocation of queries. Intuitively, structurally similar data points possess similar levels of importance, leading to a similar trend in query allocation. It is important to note that GA has a positive probability of generating negative samples. In practice, any data point receiving a negative allocation is assigned zero queries.

Bernoulli Allocator (BA) is another lightweight alternative, similar to the idea in Qin et al. (2023). In an equal allocation scenario, each data in the batch receives $\overline{A}$ queries. The BA introduces a probabilistic mechanism where, with probability $p$, the number of queries assigned to a data point is reduced to $\overline{A}/2$; otherwise, it remains at $\overline{A}$. The probability $p$ is the only parameter of the allocator. The BA is in the form of a conditional probability distribution

$$P(A_j = \overline{A}/2|\Phi = \mathcal{L}_j) = \begin{cases} p, & \mathcal{L}_j < \overline{\mathcal{L}} \\ 0, & \mathcal{L}_j \geq \overline{\mathcal{L}} \end{cases}, \tag{6}$$

where $\mathcal{L}_j$ is the clean loss on data $x_j$, and $\overline{\mathcal{L}}$ is the mean of loss value in the batch(we use the loss value without injected noise into the network).

It is important to note that $\boldsymbol{A}$ represents the relative number of queries, and thus the actual allocation proportion to the $j$-th data point is given by $A_j/\sum_{i=1}^{B} A_i$. This definition eliminates the need for imposing a fixed total budget constraint in the following allocator optimization problem. However, as a consequence, we must specify that the total query budget per step is fixed at $A_0 = \overline{A} \cdot B$ during the training process. Note that, since the data points selected in the mini-batch differ at each step, $A_j^t$ is actually a function of $t$. For simplicity of notation, we omit this dependency in the remainder of the paper.

## 4.2 OPTIMIZATION OF THE QUERY ALLOCATOR.

We employ a gradient-based method to optimize the allocator parameters $\boldsymbol{\lambda}$, introducing an additional optimization step before each iteration of neural network training. However, by formulating an equivalent optimization objective and utilizing reparameterization techniques, we ensure that this auxiliary optimization remains computationally lightweight and incurs minimal overhead.

Our objective is to maximize the performance improvement at this step, i.e.,

$$\arg\max_{\boldsymbol{\lambda} \in \Lambda} PI_t(\boldsymbol{\lambda}) \triangleq \mathbb{E}_{\boldsymbol{A} \sim P(\cdot|\boldsymbol{\lambda})}[\sum_{j=1}^{B} |\mathcal{L}_j(\theta_{t+1}) - \mathcal{L}_j(\theta_t)|], \tag{7}$$

where $\mathcal{L}_j$ is the loss corresponding to the $j$-th data point. Assuming that $\mathcal{L}_j$ is $L$-smooth, $\forall j = 1, \cdots, B$, and $\Theta$ is a convex set, we can derive a lower bound of the optimization objective $PI_t(\boldsymbol{\lambda})$.

$$PI_t(\boldsymbol{\lambda}) \geq LB_t(\boldsymbol{\lambda}) \triangleq \mathbb{E}_{\boldsymbol{A} \sim P(\cdot|\boldsymbol{\lambda})} \sum_{j=1}^{B} \left[ (\eta_t \nabla_\theta \hat{\mathcal{L}}(\theta_t))^\top \nabla_\theta \mathcal{L}_j(\theta_t) - \frac{1}{2}\eta_t^2 L \|\nabla_\theta \hat{\mathcal{L}}(\theta_t)\|^2 \right]. \tag{8}$$

Inequality (8) (the proof is provided in Appendix A.1) closely resembles the Descent Lemma commonly used in the analysis of gradient descent. The lower bound $LB_t$ is often utilized as a surrogate optimization objective in the literature on adaptive step size (Pirotta et al., 2013) and adaptive batch size (Papini et al., 2017). Since our decision variable is $\boldsymbol{\lambda}$ now, we can further simplify the form of $LB_t(\boldsymbol{\lambda})$. Before proceeding, according to the Central Limit Theorem, it is canonical to assume that the perturbation-based gradient estimator $\nabla_\theta \hat{\mathcal{L}}_j(\theta_t) = \frac{1}{A_j} \sum_{i=1}^{A_j} G(x_j, \theta_t, z_{i,j})$, $z_{i,j} \sim f(\cdot)$, $\forall j = 1, \cdots, B$, approximately follows a $d$-dimensional Gaussian distribution.

**Assumption 1.** $\forall j = 1, \cdots, B$, $\nabla_\theta \hat{\mathcal{L}}_j(\theta_t) \sim N(\mu_G^{j,t}, \frac{\Sigma_G^{j,t}}{A_j})$, where $\mu_G^{j,t}$ and $\Sigma_G^{j,t}$ are the expectation and the covariance matrix of $G(x_j, \theta_t, z)$, $z \sim f(\cdot)$, respectively.

**Theorem 1** (**Equivalent Objective**). *Suppose that the Assumption 1 holds, maximizing the lower bound $LB_t(\boldsymbol{\lambda})$ over $\boldsymbol{\lambda} \in \Lambda$ is equivalent to minimizing*

$$J_t(\boldsymbol{\lambda}) \triangleq \mathbb{E}_{\boldsymbol{A} \sim P(\cdot | \boldsymbol{\lambda})} \left[ \sum_{j=1}^{B} \mathrm{Tr}(\frac{\Sigma_G^{j,t}}{A_j}) \right] \tag{9}$$

*over $\boldsymbol{\lambda} \in \Lambda$, i.e.,*

$$\max_{\boldsymbol{\lambda} \in \Lambda} LB_t(\boldsymbol{\lambda}) \iff \min_{\boldsymbol{\lambda} \in \Lambda} J_t(\boldsymbol{\lambda}).$$

The equivalent expression (9) of the optimization objective makes the optimization of the allocator both computationally and statistically tractable (see the proof in Appendix A.2). First, it demonstrates that the optimization of the allocator is independent of both the smoothness parameter $L$ and the expectation $\mu_G^{j,t}$ of the estimator sample. Consequently, we can avoid estimating these unknown and bias-inducing high-dimensional parameters. More importantly, the optimization of the allocator **does not** require estimating the off-diagonal elements of the covariance matrix $\Sigma_G^{j,t}$, which can be statistically and computationally challenging (Fan et al., 2008), especially since $\Sigma_G^{j,t}$ is of dimension $d^2$. It only requires estimating the trace of $\Sigma_G^{j,t}$, i.e., the sum of the diagonal variances, which significantly reduces the computational complexity.

Moreover, we adopt a subsampling technique in which the trace of the covariance matrix corresponding to the parameters of the deep layers of the neural network serves as an approximation for the overall trace. This approximation is justified as these layers typically account for the majority of the variance in the model. In practice, prior to the optimization process of the allocator, we sample a small number of initial queries for each data point to estimate $\mathrm{Tr}(\Sigma_G^{j,t})$. In the subsequent computations, $\mathrm{Tr}(\Sigma_G^{j,t})$ is replaced by its estimate in a plug-in manner. In addition, from Theorem 1, we can see that generally data points with higher variance should be allocated more query resources.

Then, the gradient of $J_t(\boldsymbol{\lambda})$ with respect to allocator parameter $\boldsymbol{\lambda}$ is given by

$$\nabla_{\boldsymbol{\lambda}} J_t(\boldsymbol{\lambda}) = \nabla_{\boldsymbol{\lambda}} \mathbb{E}\left[ \sum_{j=1}^{B} \mathrm{Tr}(\frac{\Sigma_G^{j,t}}{A_j}) \right] = \mathbb{E}\left[ \sum_{j=1}^{B} \mathrm{Tr}(\frac{\Sigma_G^{j,t}}{A_j}) \nabla_{\boldsymbol{\lambda}} \ln(P(\boldsymbol{A})) \right]. \tag{10}$$

The LR gradient estimator for the allocator takes the sample average form

$$\nabla_{\boldsymbol{\lambda}} \hat{J}(\boldsymbol{\lambda}) = \frac{1}{K} \sum_{k=1}^{K} \left[ \sum_{j=1}^{B} \mathrm{Tr}(\frac{\Sigma_G^{j,t}}{A_j^{(k)}}) \nabla_{\boldsymbol{\lambda}} \ln(P(\boldsymbol{A}^{(k)})) \right], \tag{11}$$

where $\boldsymbol{A}^{(k)} = (A_1^{(k)}, \cdots, A_B^{(k)})$ are i.i.d samples from $P(\cdot | \boldsymbol{\lambda})$ and $K$ is the number of sampling from $P(\cdot | \boldsymbol{\lambda})$. At each step $t$ of neural network training, we employ the above gradient estimator to search for the optimal allocator parameter $\boldsymbol{\lambda}_t^* \triangleq \arg\max_{\boldsymbol{\lambda} \in \Lambda} J(\boldsymbol{\lambda})$. Let $LB_{1:T}^*$ and $LB_{1:T}^{\mathrm{equal}}$ represent the lower bound of the cumulative performance improvement of the neural network after $T$ steps of training when using the optimal allocator $\boldsymbol{A}^* \sim P(\cdot | \lambda_t^*)$ and using an equal allocator $\boldsymbol{A}^{\mathrm{equal}}$, respectively. Then the difference between these two allocation strategies is lower bound by the following expression.

**Theorem 2** (**Theoretical Improvement**). *Suppose that the Assumption 1 holds, then we have*

$$\mathbb{E}\left(LB_{1:T}^* - LB_{1:T}^{\mathrm{equal}}\right) \geq \sum_{t=1}^{T} \frac{\eta_t^2 L}{2BA_0} \sum_{j<k} \left( \sqrt{\mathrm{Tr}(\Sigma_G^{j,t})} - \sqrt{\mathrm{Tr}(\Sigma_G^{k,t})} \right)^2 \geq 0.$$

Theorem 2 (see the proof in Appendix A.3) quantifies the superiority of our proposed optimal allocator over the vanilla equal allocation strategy. The greater the differences in $\mathrm{Tr}(\Sigma_G^{j,t})$ across different data points (i.e., the stronger the heterogeneity), the more pronounced the superiority of the optimal allocator, and this difference equals zero if and only if all $\mathrm{Tr}(\Sigma_G^{j,t})$ are equal.

### 4.3 REPARAMETERIZATION OF THE GAUSSIAN ALLOCATOR.

The original parameter dimension of the Gaussian allocator is $B + \frac{B(B+1)}{2}$, which might be problematic due to its high dimensionality. To reduce the dimensionality of the allocator's optimization problem and to incorporate more relevant features, we adopt a reparameterization approach, i.e., imposing a specific structure to the parameter $\boldsymbol{\lambda}$.

Drawing inspiration from the literature on data pruning (Qin et al., 2023), the loss value serves as a strong indicator of the data point's characteristics. Therefore, we assume that the mean of the allocator follows a linear structure:

$$\boldsymbol{\mu} = \beta_0 + \beta_1 \Phi,$$

where $\Phi = \mathrm{Tanh}(\mathcal{L}(x; \boldsymbol{\theta}; \cdot))$ is the "clean" loss without injected noise. The original loss value is processed through a Tanh function to constraint $\Phi$ within the range $[-1, 1]$. The initial value for $\beta_0$ and $\beta_1$ are set to $\overline{A}$ and $\overline{A}/2$, respectively.

For the covariance structure $\Sigma$, based on the observation that similar data may have similar influences on the training of neural networks, we reparameterize the covariance matrix as follows: the covariance between the allocation of the $i$-th data point and the $j$-th data point is

$$\Sigma(i,j) = \sigma^2 \exp\left( - \frac{d_{i,j}}{2\gamma^2} \right),$$

where $d_{i,j}$ is the cosine similarity of the embedding between the $i$-th and the $j$-th data points. $\sigma^2$ and $\gamma^2$ are two parameters that control the scale and decay of the covariance, respectively. The $\sigma$ is initialized as $\overline{A}/5$ and $\gamma$ is initialized as 1. This kernel-based reparameterization form, also known as the Radial Basis Function (RBF), is widely used in Gaussian process regression and Bayesian Optimization literature (Shahriari et al., 2015). It enables us to model dependencies between data based on their pairwise distances.

By applying the reparameterization techniques to the covariance matrix $\Sigma$ and mean vector $\boldsymbol{\mu}$, the original high-dimensional optimization problem for these parameters has been effectively transformed into a more manageable optimization problem for 4 variables $\boldsymbol{\lambda} = (\beta_0, \beta_1, \sigma, \gamma)$. The pseudocode for the forward learning under optimal allocation is in Appendix B.

## 5 EXPERIMENTS

### 5.1 FINETUNING VISION TRANSFORMER

**Experimental setting:** We evaluate our model's performance using a diverse set of widely used benchmark datasets, each chosen for its unique characteristics and specific challenges contributing to a comprehensive analysis of our method's generalization across different domains: ImageNet (Deng et al., 2009), Caltech101 (Fei-Fei et al., 2004), Food101 (Bossard et al., 2014), Flowers102 (Nilsback & Zisserman, 2008), CIFAR10/100 (Krizhevsky et al., 2009), and EuroSAT (Helber et al., 2019).

**Evaluation metrics**: We focus on image classification and evaluate the performance by accuracy in different training contexts: BP, LR, and SPSA. We compare the performance with and without the query allocation. The GA is employed for all experiments.

Using different gradient estimators, we conduct full-parameters fine-tuning for the Vision Transformer (both base and large model). We include DeepZero (SPSA with $n$ queries, denoted as $n$-SPSA), Mezo (SPSA with only 2 queries), and LR with equally allocated query budgets as baselines in our experiments. For the SPSA-based methods, we include a version with the optimal allocator (OPS-SPSA) to compare with their corresponding baselines. We apply the same settings to the LR, comparing the optimally allocated version (OPS-LR) with the vanilla version.

We train the network for 10 epochs with the learning rate of $1 \times 10^{-4}$ and the Adam optimizer. The batch size is 128. All the methods in our experiments use the same query budgets, except for Mezo, which uses only 2 queries per data point in accordance with its original memory-efficient settings. As shown in Table 1, OPS-LR outperforms all other methods. In contrast, Mezo with only 1 query performs poorly in the experiment. On the contrary, DeepZero achieves acceptable accuracy compared with Mezo. The phenomenon suggests that an adequate number of queries is crucial for

Table 1: Classification accuracy (%) for finetuning vit on downstream dataset.

| Model | Method | ImageNet | Caltech101 | Food101 | Flowers102 | CIFAR10/100 | EuroSAT |
|---|---|---|---|---|---|---|---|
| ViT-base | BP (upper bound) | 80.5 | 92.2 | 92.4 | 93.5 | 96.3 / 90.7 | 91.9 |
| | Mezo (2 queries) | 9.3 | 42.6 | 37.3 | 38.2 | 42.1 / 37.5 | 39.8 |
| | Mezo ($n$-SPSA) | 54.8 | 77.2 | 76.7 | 80.2 | 84.4 / 71.5 | 78.4 |
| | DeepZero ($n$-SPSA) | 55.6 | 78.4 | 77.8 | 78.4 | 85.6 / 70.2 | 78.8 |
| | LR | 57.3 | 85.5 | 87.6 | 86.5 | 87.7 / 80.8 | 82.9 |
| | OPS-SPSA (Ours) | 60.8 | 87.3 | 87.6 | 85.6 | 89.5 / 82.2 | 83.2 |
| | OPS-LR (Ours) | **65.8** | **88.9** | **88.4** | **91.6** | **93.2 / 88.7** | **89.3** |
| ViT-Large | BP (upper bound) | 81.7 | 93.4 | 93.3 | 94.2 | 98.2 / 93.5 | 93.4 |
| | Mezo (2 queries) | 10.7 | 45.6 | 38.8 | 40.5 | 45.9 / 38.3 | 41.7 |
| | Mezo ($n$-SPSA) | 56.7 | 78.8 | 78.4 | 78.5 | 86.7 / 74.3 | 79.2 |
| | DeepZero ($n$-SPSA) | 56.4 | 79.5 | 78.9 | 79.6 | 86.3 / 73.7 | 80.1 |
| | LR | 58.2 | 87.2 | 87.7 | 86.8 | 88.4 / 81.4 | 85.7 |
| | OPS-SPSA (Ours) | 62.2 | 88.6 | 88.1 | 87.3 | 91.3 / 83.9 | 87.6 |
| | OPS-LR (Ours) | **67.6** | **90.2** | **89.5** | **92.3** | **95.5 / 90.8** | **90.1** |

the success of perturbation-based methods since the gradient's variance must be properly controlled. The OPS-SPSA achieves better accuracy than the DeepZero with an equally allocated budget. Our proposed optimal allocator provides universal acceleration across all perturbation-based methods. Moreover, LR achieves higher accuracy than DeepZero, suggesting that LR is a better gradient estimator than the SPSA in this experiment context. Please refer to the Appendix C.1 for the details and further analysis.

## 5.2 BLACK-BOX TUNING FOR MODEL AS A SERVICE

We can tune the prompts for vision-language models (VLMs) in a Model-as-a-Service (MaaS) environment using the OPS-LR framework, as shown in Figure 2. We address the challenge of fine-tuning VLMs without access to model internals or gradient information through a scalable black-box optimization approach. We use CLIP (Radford et al., 2021), with embedding dimensions of $D_T = 512$ for the text encoder and $D_I = 756$ for the image encoder, as the backbone foundation model. Both the text encoder and the image encoder are kept frozen as black-box modules.

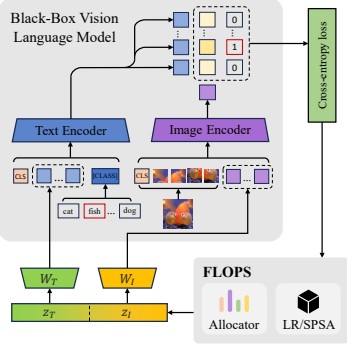

Tunable prefix or suffix prompt tokens are injected into the text and image token sequences, respectively: $S_T = [cls, p_T, e_T] \in \mathbb{R}^{(1+n_T+m_T) \times D_T}$ is the input sequence for the text encoder, $S_I = [cls, e_I, p_I] \in \mathbb{R}^{(1+m_I+n_I) \times D_I}$ for the image counterpart. Previous researches have suggested that large foundation models have low intrinsic dimensions.

Figure 2: illustration paradigm of fine-tuning the prompt for black-box vision language model.

Therefore, it is plausible to project the original embedding space onto a low-dimension subspace: $p_T = z_T \cdot W_T; W_T \in \mathbb{R}^{d_T \times D_T}, z_T \in \mathbb{R}^{m_T \times d_T}$ and $p_I = z_I \cdot W_I; W_I \in \mathbb{R}^{d_I \times D_I}, z_I \in \mathbb{R}^{m_I \times d_I}$. $W_T$ and $W_I$ are the projection matrix. The projection matrix is kept frozen after being initialized with a Gaussian distribution. With the projection, our black-box tuning problem is reduced to dimensions of $d_T + d_I \ll D_T + D_I$. The optimization problem, using the cross-entropy loss, takes the form:

$$Z^* = \arg\max_{z_T, z_I} \mathcal{L}([e_T, e_I]; \boldsymbol{\theta} = [z_T \cdot D_T, z_I \cdot D_I]; \cdot), \tag{12}$$

where $Z$ is the unified formulation of $[z_T, z_I]$. By applying the OPS-LR framework, we can optimize the black-box prompt tuning problems without access to the model structure. We can achieve an accuracy of 69.7% on ImageNet by tuning only a few thousand parameters, surpassing the performance of other black-box methods and Manual Prompting (Radford et al., 2021). The performance is shown in Table 2. Please refer to the Appendix C.2 for more details.

## 5.3 ALIGNMENT BETWEEN BLACK-BOX FOUNDATION MODEL

Foundation models like GPT (Brown, 2020), LLaMa Touvron et al. (2023), Vicuna (Chiang et al., 2023), and CLIP (Radford et al., 2021) contain rich domain knowledge from the pretraining. Multimodal tasks, such as video captioning (Krishna et al., 2017) and video grounding(Gao et al., 2017), require models to have sufficient cross-domain understanding, and proper alignment between different modalities is essential.

Table 2: Classification accuracy (%) for black-box prompt tuning.

| Methods | ImageNet | Caltech101 | Food101 |
|---|---|---|---|
| OPS-LR | **65.1** | **90.5** | **82.7** |
| OPS-SPSA | 64.2 | 89.3 | 81.5 |
| LR | 62.5 | 87.4 | 79.2 |
| SPSA | 60.7 | 85.3 | 77.6 |
| Manual Prompt | 57.9 | 83.3 | 76.4 |

Aligning foundation models from a single modality is a more economical approach than pretraining from scratch using multimodal data. It is a common practice to add an adapter between the visual encoder and the Large Language model. The adapter is a linear projector, $g(\cdot)$, that projects the visual embedding to the textual embedding space. Since foundation models contain billions of parameters, the backward pass through the model requires extra memory for storing the computation graph, and recursive computations through such a large pre-trained model are time-consuming. Our OPS-LR framework efficiently bypasses the need for recursive computation to estimate the gradient with a large foundation model in between, as shown in Figure 3.

The training paradigm for the multimodal alignment follows an autoregressive style using image-text pairs $< I, T >$. We use a frozen CLIP ViT-L/14 as the visual encoder, denoted as $ViT$. The CLS tokens of the output sequence of $ViT$ are utilized as the global feature for the frame. We denote the visual token after the projection as $Z_v = g(ViT(I)^{CLS})$. The visual token is then concatenated with the text tokens sequence: $[Z_v, t_1, t_2, ..., t_m]$, where $[t_1, t_2, ..., t_m]$ is the text tokens sequence with a length of $m$. Consequently, we can train the adapter on the concatenated sequence with the autoregressive objective of the LLM. We report the wall time for the feature alignment procedure on the LCS-558K dataset (Liu et al., 2024). Our OPS-LR framework can reduce almost half of the training time since the recursive backward computation is skipped. From Figure 5, the wall time with or without the allocator is almost the same, which further illustrates that our allocator is lightweight and has minimal impact on runtime. Please refer to the Appendix C.3 for more details.

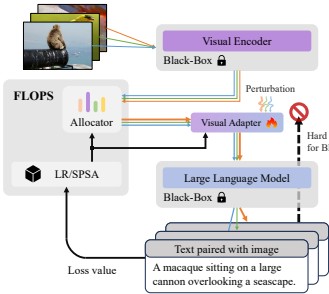

Figure 3: illustration alignment between foundation model for video understanding.

## 5.4 ABLATION STUDY

We conduct an ablation study using ViT-base on the CIFAR100 dataset. We investigate the necessity of the allocation policy for the forward-learning training and determine whether GA or BA is the better allocator. Furthermore, we discern the varying degrees of difficulty in estimation across different network components. The results are shown in Figure 4.

**Impact of different query budgets.** To further investigate the necessity of the procedure of query budget allocation, we compare the performance increment under various query budgets for different allocation policies. Traditionally, the queries are equally allocated (EA) across different data. The performance improvements offered by GA and BA are considerably more significant than those of EA. When the number of queries per data is small, the benefit from the allocation is still non-ignorable. In other words, the more query budget you have, the greater the improvement gains from allocation. Meanwhile, the ablation on different transformer layers indicates that the variance of the gradient increases as the layer goes deeper, which validates our sub-sampling technique.

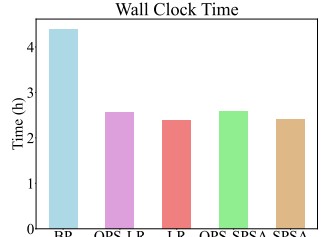

Figure 5: Wall clock time for feature alignment between different methods.

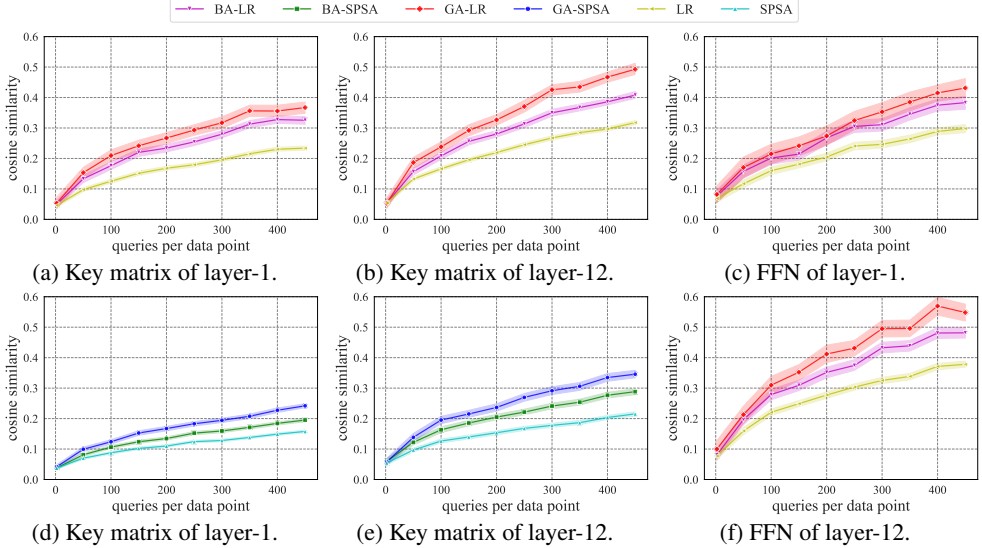

Figure 4: Ablation study on the effect of different allocators and estimation difficulty at different layers. We show the cosine similarity between the estimated and true gradients. We estimate the gradient of the Key matrix in multi-head attention (MHA) and the first linear layer in the feed-forward network (FFN)). Layer 1 is adjacent to the embedding and the layer 12 is adjacent to the classification head.

**Impact of different allocators, Gaussian or Bernoulli.** We compare different allocators in Figures 4(a), (b), (d), and (e). GA controls the allocation with the mean vectors and the covariance matrix. The covariance matrix captures the similarity between different data points. Structurally similar data points possess similar levels of importance. Therefore, the allocation policy for those data points should be correlated. If the covariance matrix is diagonal, the similarity between data points is ignored, which could undermine the performance of the allocation strategy. Compared with the GA, the BA is a simpler strategy. Intuitively, it prunes the queries on unimportant data points to eliminate unnecessary computation. Our ablation study suggests that GA is superior to BA.

**Estimation on different components of Transformer.** We present the cosine similarity results on the multi-head attention (MHA) and the feed-forward network (FFN) in Figures 4(a)-(c) and (f). Estimation on the FFN is easier than on the MHA. In the Transformer block, the MHA precedes the FFN, which may explain this phenomenon. The low cosine similarity on MHA attention of deep layers may lead to the performance gap on large-scale dataset like ImageNet. It is also worth noting that a large number of queries when training Transformer can improve the estimation quality. However, the memory cost of increasing queries is extremely high since the complexity of the attention module is $\mathcal{O}(n^2)$. Integrating efficient MHA techniques, such as Flash Attention (Dao et al., 2022), is left for future work.

## 6 CONCLUSIONS

In this paper, we present a unified perspective on forward learning through a perturbation-based framework and propose an optimal query allocation strategy to enhance its efficiency. By leveraging a likelihood ratio-based objective and a reparameterized Gaussian allocator, our method adaptively distributes queries based on data-specific gradient estimation difficulty, ensuring an optimal balance between computational cost and estimation accuracy. Theoretical analysis guarantees the optimality of our allocation strategy, while extensive experiments demonstrate its effectiveness in fine-tuning Vision Transformers and addressing real-world black-box optimization problems, such as prompt tuning and multimodal model alignment. Our results show that the proposed method significantly improves the scalability of forward learning, achieving state-of-the-art performance compared to existing approaches. While our method significantly improves scalability, closing the gap with back-propagation remains an open challenge for future research.

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

## A  THEORETICAL DETAILS

### A.1  PROOF OF INEQUALITY (8)

$$PI_t(\boldsymbol{\lambda}) \geq \mathbb{E}_{\boldsymbol{A} \sim P(\cdot|\boldsymbol{\lambda})} \sum_{j=1}^{B} \left[ (\eta_t \nabla_\theta \hat{\mathcal{L}}(\theta_t))^\top \nabla_\theta \mathcal{L}_j(\theta_t) - \frac{1}{2} \eta_t^2 L \|\nabla_\theta \hat{\mathcal{L}}(\theta_t)\|^2 \right].$$

*Proof.* For each loss function $\mathcal{L}_j$, by exploiting Taylor's theorem, we have

$$
\begin{aligned}
\mathcal{L}_j(\theta_{t+1}) - \mathcal{L}_j(\theta_t) &= \int_0^1 \frac{d}{d\tau} \mathcal{L}_j(\theta_t + \tau(\theta_{t+1} - \theta_t)) \, d\tau \\
&= \int_0^1 \langle \nabla_\theta \mathcal{L}_j(\theta_t + \tau(\theta_{t+1} - \theta_t)), \theta_{t+1} - \theta_t \rangle \, d\tau.
\end{aligned}
\tag{13}
$$

By adding and subtracting $\nabla_\theta \mathcal{L}_j(\theta_t)$,

$$
\begin{aligned}
(13) &= \int_0^1 \langle \nabla_\theta \mathcal{L}_j(\theta_t), \theta_{t+1} - \theta_t \rangle \, d\tau + \int_0^1 \langle \nabla_\theta \mathcal{L}_j(\theta_t + \tau(\theta_{t+1} - \theta_t)) - \nabla_\theta \mathcal{L}_j(\theta_t), \theta_{t+1} - \theta_t \rangle \, d\tau \\
&= \langle \nabla_\theta \mathcal{L}_j(\theta_t), \theta_{t+1} - \theta_t \rangle + \int_0^1 \langle \nabla_\theta \mathcal{L}_j(\theta_t + \tau(\theta_{t+1} - \theta_t)) - \nabla_\theta \mathcal{L}_j(\theta_t), \theta_{t+1} - \theta_t \rangle \, d\tau.
\end{aligned}
\tag{14}
$$

Now we focus on the second term in (14). By the Cauchy-Schwarz inequality, we have

$$|\langle \nabla_\theta \mathcal{L}_j(\theta_t + \tau(\theta_{t+1} - \theta_t)) - \nabla_\theta \mathcal{L}_j(\theta_t), \theta_{t+1} - \theta_t \rangle| \leq \|\nabla_\theta \mathcal{L}_j(\theta_t + \tau(\theta_{t+1} - \theta_t)) - \nabla_\theta \mathcal{L}_j(\theta_t)\| \cdot \|\theta_{t+1} - \theta_t\|.$$

And according to the $L$-smoothness assumption, we have

$$\|\nabla_\theta \mathcal{L}_j(\theta_t + \tau(\theta_{t+1} - \theta_t)) - \nabla_\theta \mathcal{L}_j(\theta_t)\| \leq L\|\tau(\theta_{t+1} - \theta_t)\| = L\tau\|\theta_{t+1} - \theta_t\|.$$

Therefore, the second term in (14) becomes

$$
\begin{aligned}
\int_0^1 &\langle \nabla_\theta \mathcal{L}_j(\theta_t + \tau(\theta_{t+1} - \theta_t)) - \nabla_\theta \mathcal{L}_j(\theta_t), \theta_{t+1} - \theta_t \rangle \, d\tau \\
&\geq -\int_0^1 L\tau\|\theta_{t+1} - \theta_t\|^2 \, d\tau = -\frac{L}{2}\|\theta_{t+1} - \theta_t\|^2.
\end{aligned}
\tag{15}
$$

Combining (14) and (15), and substitute $\theta_{t+1} = \theta_t + \eta_t \nabla_\theta \hat{\mathcal{L}}(\theta_t)$, we have

$$\mathcal{L}_j(\theta_t + \eta_t \nabla_\theta \hat{\mathcal{L}}(\theta_t)) - \mathcal{L}_j(\theta_t) \geq \eta_t \langle \nabla_\theta \mathcal{L}_j(\theta_t), \nabla_\theta \hat{\mathcal{L}}(\theta_t) \rangle - \frac{L}{2} \eta_t^2 \|\nabla_\theta \hat{\mathcal{L}}(\theta_t)\|^2.$$

Then, taking expectation over $\boldsymbol{A} \sim P(\cdot|\boldsymbol{\lambda})$ and summing over $j$,

$$PI_t(\boldsymbol{\lambda}) \geq \mathbb{E}_{\boldsymbol{A} \sim P(\cdot|\boldsymbol{\lambda})} \sum_{j=1}^{B} \left[ (\eta_t \nabla_\theta \hat{\mathcal{L}}(\theta_t))^\top \nabla_\theta \mathcal{L}_j(\theta_t) - \frac{1}{2} \eta_t^2 L \|\nabla_\theta \hat{\mathcal{L}}(\theta_t)\|^2 \right],$$

which completes the proof. $\qquad \square$

### A.2  PROOF OF THEOREM 1

*Proof.*

$$
\begin{aligned}
LB_t &= \mathbb{E} \sum_{j=1}^{B} \left[ (\eta_t \nabla_\theta \hat{\mathcal{L}}(\theta_t))^\top \nabla_\theta \mathcal{L}_j(\theta_t) - \frac{1}{2} \eta_t^2 L \|\nabla_\theta \hat{\mathcal{L}}(\theta_t)\|^2 \right] \\
&= \mathbb{E} \sum_{j=1}^{B} \left[ \left( \frac{\eta_t}{B} \sum_{j=1}^{B} \nabla_\theta \hat{\mathcal{L}}_j(\theta_t) \right)^\top \nabla_\theta \mathcal{L}_j(\theta_t) - \frac{1}{2} \eta_t^2 L \| \frac{1}{B} \sum_{j=1}^{B} \nabla_\theta \hat{\mathcal{L}}_j(\theta_t) \|^2 \right],
\end{aligned}
\tag{16}
$$

where $\nabla_\theta \hat{\mathcal{L}}_j(\theta_t) \sim N(\mu_G^{j,t}, \frac{\Sigma_G^{j,t}}{A_j})$, and $j = 1, \cdots, B$. Then, by taking the conditional expectation.

$$LB_t = \mathbb{E}_{\boldsymbol{A}} \sum_{j=1}^{B} \left[ \mathbb{E}_{\nabla_\theta \hat{\mathcal{L}}(\theta_t)} \left[ \left( \frac{\eta_t}{B} \sum_{j=1}^{B} \nabla_\theta \hat{\mathcal{L}}_j(\theta_t) \right)^\top \nabla_\theta \mathcal{L}_j(\theta_t) - \frac{1}{2} \eta_t^2 L \left\| \frac{1}{B} \sum_{j=1}^{B} \nabla_\theta \hat{\mathcal{L}}_j(\theta_t) \right\|^2 \Big| \boldsymbol{A} \right] \right] \tag{17}$$

$\forall j = 1, \cdots, B$, the expected inner product on the LHS of (17)

$$\mathbb{E}_{\nabla_\theta \hat{\mathcal{L}}(\theta_t)} \left[ \left( \frac{\eta_t}{B} \sum_{j=1}^{B} \nabla_\theta \hat{\mathcal{L}}_j(\theta_t) \right)^\top \nabla_\theta \mathcal{L}_j(\theta_t) \Big| \boldsymbol{A} \right] = \frac{\eta_t}{B} \left( \sum_{j=1}^{B} \mu_G^{j,t} \right)^\top \nabla_\theta \mathcal{L}_j(\theta_t). \tag{18}$$

Next, we focus on the expected squared norm on the RHS of (17).

$$\mathbb{E}_{\nabla_\theta \hat{\mathcal{L}}(\theta_t)} \left[ \left\| \frac{1}{B} \sum_{j=1}^{B} \nabla_\theta \hat{\mathcal{L}}_j(\theta_t) \right\|^2 \Big| \boldsymbol{A} \right] = \frac{1}{B^2} \left( \sum_{j=1}^{B} \sum_{k=1}^{B} \mathbb{E} \left[ \nabla_\theta \hat{\mathcal{L}}_j(\theta_t)^\top \nabla_\theta \hat{\mathcal{L}}_k(\theta_t) \Big| \boldsymbol{A} \right] \right). \tag{19}$$

For $j \neq k$, we have
$$\mathbb{E} \left[ \nabla_\theta \hat{\mathcal{L}}_j(\theta_t)^\top \nabla_\theta \hat{\mathcal{L}}_k(\theta_t) \Big| \boldsymbol{A} \right] = \left( \mu_G^{j,t} \right)^\top \mu_G^{k,t}. \tag{20}$$

For $j = k$, we express $\nabla_\theta \hat{\mathcal{L}}_j(\theta_t)$ as the sum of the mean and a zero-mean normal random vector:
$$\nabla_\theta \hat{\mathcal{L}}_j(\theta_t) = \mu_G^{j,t} + Z,$$
where $Z \sim N(0, \frac{\Sigma_G^{j,t}}{A_j})$, meaning $Z$ is a zero-mean random vector with covariance $\frac{\Sigma_G^{j,t}}{A_j}$. Then we have

$$\begin{aligned} \mathbb{E} \left[ \nabla_\theta \hat{\mathcal{L}}_j(\theta_t)^\top \nabla_\theta \hat{\mathcal{L}}_k(\theta_t) \Big| \boldsymbol{A} \right] &= \mathbb{E} \left[ \left( \mu_G^{j,t} + Z \right)^\top \left( \mu_G^{j,t} + Z \right) \Big| \boldsymbol{A} \right] \\ &= \mathbb{E} \left[ \left( \mu_G^{j,t} \right)^\top \mu_G^{j,t} + 2 \left( \mu_G^{j,t} \right)^\top Z + Z^\top Z \Big| \boldsymbol{A} \right] \\ &= \left\| \mu_G^{j,t} \right\|^2 + 0 + \mathbb{E} \left[ Z^\top Z | A \right]. \end{aligned} \tag{21}$$

The expected value of the quadratic form $Z^\top Z$ in (21) for $Z \sim N(0, \frac{\Sigma_G^{j,t}}{A_j})$ is given by the trace of the covariance matrix, i.e.,

$$\mathbb{E} \left[ Z^\top Z \right] = \mathrm{Tr} \left( \frac{\Sigma_G^{j,t}}{A_j} \right). \tag{22}$$

Therefore, with (20), (21) and (22), we have

$$(19) = \frac{1}{B^2} \left( \left\| \sum_{j=1}^{B} \mu_G^{j,t} \right\|^2 + \sum_{j=1}^{B} \mathrm{Tr} \left( \frac{\Sigma_G^{j,t}}{A_j} \right) \right). \tag{23}$$

Then, with (17), (18) and (23), we have

$$\begin{aligned} LB_t(\boldsymbol{\lambda}) = \mathbb{E}_{\boldsymbol{A} \sim p(\cdot|\boldsymbol{\lambda})} \Bigg[ &\frac{\eta_t}{B} \left( \sum_{j=1}^{B} \mu_G^{j,t} \right)^\top \left( \sum_{j=1}^{B} \nabla_\theta \mathcal{L}_j(\theta_t) \right) \\ &- \frac{1}{2} \sum_{j=1}^{B} \frac{\eta_t^2 L}{B^2} \left( \left\| \sum_{j=1}^{B} \mu_G^{j,t} \right\|^2 + \sum_{j=1}^{B} \mathrm{Tr} \left( \frac{\Sigma_G^{j,t}}{A_j} \right) \right) \Bigg]. \end{aligned} \tag{24}$$

Notice that, $\mu_G^{j,t}$ is independent of the allocation $\boldsymbol{A}$, which only depends on the estimator type and the parameters of the injected noise in the perturbation. Therefore, it is easy to see that maximizing the lower bound $LB_t(\boldsymbol{\lambda})$ over $\boldsymbol{\lambda} \in \Lambda$ is equivalent to minimizing

$$J_t(\boldsymbol{\lambda}) \triangleq \mathbb{E}_{\boldsymbol{A} \sim P(\cdot|\boldsymbol{\lambda})} \left[ \sum_{j=1}^{B} \mathrm{Tr} \left( \frac{\Sigma_G^{j,t}}{A_j} \right) \right] \tag{25}$$

over $\boldsymbol{\lambda} \in \Lambda$. $\qquad \square$

## A.3 PROOF OF THEOREM 2

*Proof.* Based on the optimality of $\lambda_t^*$ at step $t$, we know that for any GA following $N(\mu, \Sigma)$, the corresponding $\mathbb{E}(LB_t^{GA}) \leq \mathbb{E}(LB_t^*)$. Now, if we set $\Sigma = \mathbf{0}_{B \times B}$, the Gaussian distribution becomes deterministic, and the query allocation becomes a fixed, non-random strategy. To establish a lower bound for $\mathbb{E}(LB_t^*) - \mathbb{E}(LB_t^{\text{equal}})$, we construct an optimal deterministic allocator $\mathbf{A}^{\text{det}}$ as the intermediary strategy between $\mathbf{A}^*$ and $\mathbf{A}^{\text{equal}}$, which satisfies

$$\mathbb{E}(LB_t^{\text{equal}}) \leq \mathbb{E}(LB_t^{\text{det}}) \leq \mathbb{E}(LB_t^*),$$

where $LB_t^{\text{det}}$ is lower bound of the PI corresponding to $\mathbf{A}^{\text{det}}$.

For the optimal deterministic allocator $\mathbf{A}^{\text{det}}$, we aim to maximize the objective function

$$J_t(\mathbf{A}) = \sum_{j=1}^{B} \frac{\text{Tr}(\Sigma_G^{j,t})}{A_j}$$

subject to the constraint $\sum_{j=1}^{B} A_j = A_0$. The Lagrangian function is

$$L = \sum_{j=1}^{B} \frac{\text{Tr}(\Sigma_G^{j,t})}{A_j} - \lambda \left( \sum_{j=1}^{B} A_j - A_0 \right),$$

where $\lambda$ is the Lagrange multiplier. Take the partial derivative of $L$ with respect to each $A_j$ and set it to zero.

$$\frac{\partial L}{\partial A_j} = -\frac{\text{Tr}(\Sigma_G^{j,t})}{A_j^2} - \lambda = 0.$$

Then we have

$$A_j^{\text{det}} = A_0 \cdot \frac{\sqrt{\text{Tr}(\Sigma_G^{j,t})}}{\sum_{k=1}^{B} \sqrt{\text{Tr}(\Sigma_G^{k,t})}}.$$

Then corresponding objective is

$$J_t(\mathbf{A}^{\text{det}}) = \sum_{j=1}^{B} \frac{\text{Tr}(\Sigma_G^{j,t})}{A_j} = \sum_{j=1}^{B} \frac{\text{Tr}(\Sigma_G^{j,t})}{A_0 \cdot \frac{\sqrt{\text{Tr}(\Sigma_G^{j,t})}}{\sum_{k=1}^{B} \sqrt{\text{Tr}(\Sigma_G^{k,t})}}} = \frac{\left( \sum_{j=1}^{B} \sqrt{\text{Tr}(\Sigma_G^{j,t})} \right)^2}{A_0}.$$

As for the equal allocator $A_j^{\text{equal}} = \frac{A_0}{B}$, the corresponding objective function becomes

$$J_t(\mathbf{A}^{\text{equal}}) = \frac{A_0}{B} = \sum_{j=1}^{B} \frac{\text{Tr}(\Sigma_G^{j,t})}{\frac{A_0}{B}} = \frac{B}{A_0} \sum_{j=1}^{B} \text{Tr}(\Sigma_G^{j,t}).$$

It follows that

$$J_t(\mathbf{A}^{\text{equal}}) - J_t(\mathbf{A}^{\text{det}}) = \frac{\left( \sum_{j=1}^{B} \sqrt{\text{Tr}(\Sigma_G^{j,t})} \right)^2}{A_0} - \frac{B}{A_0} \sum_{j=1}^{B} \text{Tr}(\Sigma_G^{j,t}).$$

Notice that

$$B \sum_{j=1}^{B} \text{Tr}(\Sigma_G^{j,t}) - \left( \sum_{j=1}^{B} \sqrt{\text{Tr}(\Sigma_G^{j,t})} \right)^2 = \sum_{j<k} \left( \sqrt{\text{Tr}(\Sigma_G^{j,t})} - \sqrt{\text{Tr}(\Sigma_G^{k,t})} \right)^2.$$

Then

$$J_t(\mathbf{A}^{\text{equal}}) - J_t(\mathbf{A}^{\text{det}}) = \frac{1}{A_0} \sum_{j<k} \left( \sqrt{\text{Tr}(\Sigma_G^{j,t})} - \sqrt{\text{Tr}(\Sigma_G^{k,t})} \right)^2.$$

This difference is always non-negative and equals zero if and only if all $\mathrm{Tr}(\Sigma_G^{j,t})$ are equal. Then we have

$$
\begin{aligned}
\mathbb{E}\big(LB_{1:T}^* - LB_{1:T}^{\text{equal}}\big) \geq \mathbb{E}\big(LB_{1:T}^{\text{det}} - LB_{1:T}^{\text{equal}}\big) &= \sum_{t=1}^{T} \frac{\eta_t^2 L}{2B}(J_t(\boldsymbol{A}^{\text{equal}}) - J_t(\boldsymbol{A}^{\text{det}})) \\
&\geq \sum_{t=1}^{T} \frac{\eta_t^2 L}{2B A_0} \sum_{j<k}\left(\sqrt{\mathrm{Tr}(\Sigma_G^{j,t})} - \sqrt{\mathrm{Tr}(\Sigma_G^{k,t})}\right)^2,
\end{aligned}
\tag{26}
$$

which completes the proof. $\qquad\square$

## B  PSEUDOCODE

---

**Algorithm 1** Perturbation-based training via optimal allocation

---

**Input:** Target module $\varphi(\cdot;\theta)$, loss function $\mathcal{L}(\cdot)$, dataset $\mathcal{X}$, noise density $f(\cdot)$, allocator $P(\boldsymbol{A}|\boldsymbol{\lambda},\Phi)$, update interval $M$.

1: Initialize network parameter $\theta$ and allocator parameter $\boldsymbol{\lambda}$.
2: **repeat**
3:     $t \leftarrow 0$.
4:     **for** one mini-batch $\{x_i\}_{i=0}^{B-1}$ non-overlapping **sampled in** $\mathcal{X}$ **do**
5:         Compute the loss, $\Phi = \mathcal{L}(y)$, without injected noise.
6:         Sample initial queries to estimate $\mathrm{Tr}(\Sigma_G^{j,t})$.
7:         **repeat**
8:             Update the $\lambda$ by estimated gradient following (11).
9:         **until** $\lambda$ converges
10:         Sample the allocation decision $\boldsymbol{A} \sim P(\boldsymbol{A}|\boldsymbol{\lambda},\Phi)$ for the mini-batch.
11:         Augment the $\{x_i\}_{i=0}^{B-1}$ according to $\boldsymbol{A} = (A_1, A_2, ..., A_B)$.
12:         **for** $\theta_j \in \theta$ **do**
13:             Compute the $\mathcal{L}(y|_z)$ with $z \sim f(\cdot)$ injected to the $\varphi(\cdot;\theta_j)$.
14:             Update $\theta_j$ by estimated gradient following (1) or (3) with $\boldsymbol{A}$ queries.
15:         **end for**
16:         $t \leftarrow t + 1$.
17:     **end for**
18: **until** network parameter converges.

**Output:** Network parameter $\theta$.

---

## C  EXPERIMENT DETAILS

**Platform**: All the experiments are conducted on a machine with 8 NVIDIA A800 GPUs. Each A800 GPU has 80GB of memory.

### C.1  EXPERIMENTS FOR VIT

**Datasets**: **ImageNet:** Over 1.2 million images in 1,000 classes, a standard large-scale challenge for object classification. **Caltech101:** 9,144 images across 101 categories, focusing on object recognition with varying perspectives and poses. **Food101:** 101,000 images in 101 food categories, presenting challenges in texture, lighting, and fine-grained classification. **Flower102:** 8,189 images across 102 flower species, testing subtle feature recognition. **CIFAR-10/100:** Consisting of 60,000 32x32 images, CIFAR-10 has 10 classes, while CIFAR-100 has 100 classes, offering challenges in small-scale image classification with increasing complexity in category distinctions. **EuroSAT:** 27,000 satellite images in 10 land-use categories, challenging models to perform in remote sensing tasks. All the input images are resized to 224 before entering the ViT backbone.

**ViT-base**: The base model has 12 transformer layers. The hidden dimension is 786, with 12 heads for the multi-head attention. The intermediate dimension for feed-forward mlp is 3072. 12 layers

are equally divided into three groups from bottom to top. The standard deviation of the injected Gaussian noise for each group is initialized as $1 \times 10^{-5}, 1 \times 10^{-4}, 1 \times 10^{-3}$. The query budget for each group is 20 queries per data. The query budget for the classification head is also 20 queries per data.

**ViT-large**: The large model has 24 transformer layers. The hidden dimension is 1024, with 16 heads for the multi-head attention. The intermediate dimension for feed-forward mlp is 4096. 24 layers are equally divided into four groups from bottom to top. The standard deviation of the injected Gaussian noise for each group is initialized as $5 \times 10^{-6}, 1 \times 10^{-5}, 1 \times 10^{-4}, 1 \times 10^{-3}$. The query budget for each group is 20 queries per data. The query budget for the classification head is 20 queries per data.

We use the CLS token on top of the Transformer backbone to calculate the cosine similarity between different data points, $d_{i,j}$, in the kernel for reparameterization. The pruned queries for the BA will be equally allocator to the other unpruned data points to ensure the total number of queries is the same.

Table 3: Results of ViT-base under natural noise and adversarial noise.

| Datasets | Methods | Original | Natural Noise | | | Adversarial Noise | | |
|---|---|---|---|---|---|---|---|---|
| | | | Gaussian | Uniform | Poisson | FGSM | I-FGSM | MI-FGSM |
| ImageNet | BP | 80.5 | 78.4 | 65.9 | 44.2 | 15.2 | 10.9 | 9.9 |
| | OPS-LR | 65.8 | 65.4 | 65.4 | 60.3 | 34.5 | 15.6 | 13.8 |
| | OPS-SPSA | 60.8 | 58.2 | 59.3 | 55.2 | 33.7 | 13.5 | 12.3 |
| Food101 | BP | 92.4 | 88.3 | 84.7 | 66.4 | 19.6 | 12.3 | 10.5 |
| | OPS-LR | 88.4 | 85.3 | 82.1 | 70.3 | 37.2 | 25.2 | 18.6 |
| | OPS-SPSA | 87.6 | 84.7 | 80.3 | 70.4 | 36.4 | 23.6 | 15.3 |
| CIFAR100 | BP | 90.7 | 83.8 | 75.2 | 43.5 | 18.3 | 9.3 | 7.5 |
| | OPS-LR | 88.7 | 84.3 | 77.6 | 52.9 | 25.4 | 14.8 | 12.3 |
| | OPS-SPSA | 82.2 | 79.3 | 76.8 | 48.2 | 22.5 | 13.8 | 9.4 |

**Evaluation of robustness**: Furthermore, we assess the robustness of BP, OPS-LR, and OPS-SPSA across three data sets—ImageNet, Food101, and CIFAR100—through three distinct criteria: 1) Primary task efficacy: accuracy of classification on unaltered samples (Original); 2) Endurance against natural corruption: accuracy of classification on datasets tainted with natural noise, incorporating Gaussian, uniform, and Poisson disturbances; 3) Robustness to adversarial attacks: accuracy of classification on samples altered by adversarial tactics, specifically the fast gradient sign method (FGSM), iterative fast gradient sign method (I-FGSM), and momentum-based iterative fast gradient sign method (MI-FGSM). For adversarial assaults, we cap the allowable perturbation per pixel at $8/255$, and for I-FGSM and MI-FGSM, we limit the iteration count to a maximum of 5. In our evaluations, we encompass the entire test set for corruption assessment. As shown in Table 3, BP achieves the best accuracy on the original data. However, the robustness of OPS-LR and OPS-SPSA is superior to BP. Especially under adversarial attacks, the perturbation methods have a significant advantage over BP.

## C.2 EXPERIMENTS FOR BLACK-BOX TUNING

We use the open-source CLIP with ViT-B/32 as the visual encoder. The intrinsic dimension, $d_I + d_T$, is 1000. The visual prompt length is 10 and the text prompt length is 12. The batch size is 64. We use 60 queries per data to tune the intrinsic dimension. We use Adam optimizer with a learning rate of $1 \times 10^{-4}$ and train for 50 epochs with early stopping. All methods use the same 16-shot split for training and are evaluated on the full test sets for evaluation.

## C.3 EXPERIMENTS FOR MULTIMODAL ALIGNMENT

we use Vicuna v1.5 7B as the Large Language Model and train the 7B version. We train one epoch for the feature alignment by an Adam optimizer with the learning rate of $5 \times 10^{-4}$ and cosine learning rate decay. The batch size is 64 and the query budget is 60 queries per data.

## C.4 EXTENDED ABLATION

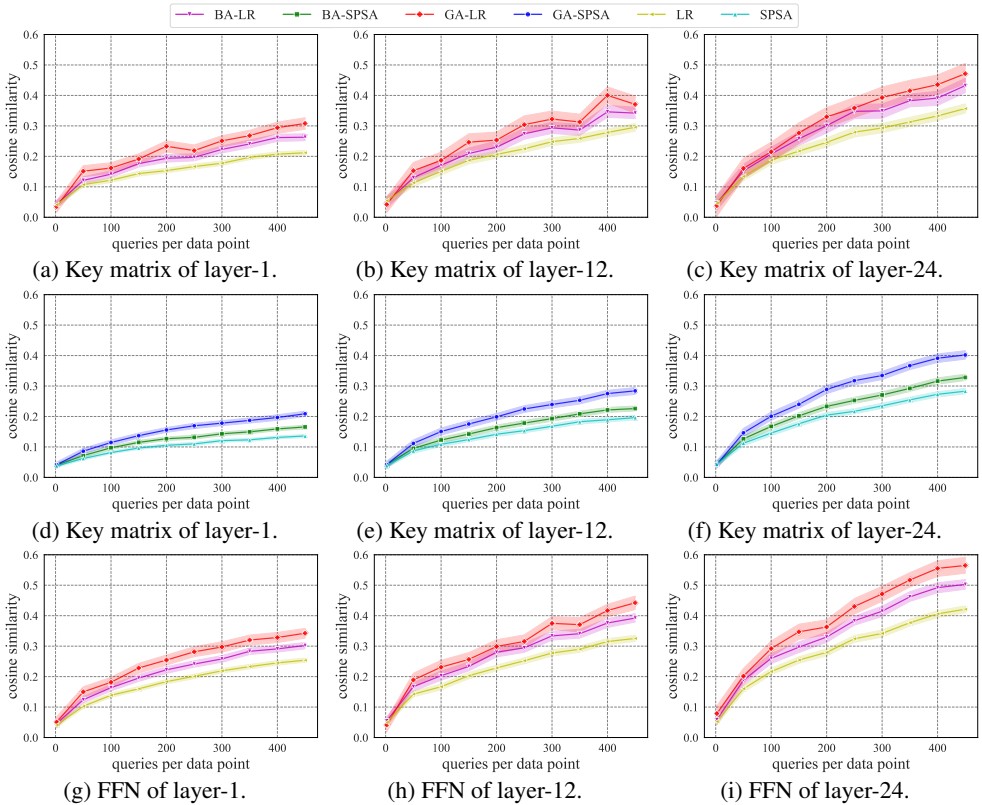

Figure 6: We show the cosine similarity between the estimated and true gradients. We compare LR and SPSA estimators on the Key matrix of the multi-head attention. The estimation of the feed-forward network is also included. Layer 1 is the Transformers layer adjacent to the embedding and layer 24 is adjacent to the classification head.

We provide a more extensive ablation study on ViT-Large following the same setting as the main text. Figures 6(a)-(c) and (d)-(f) show the result of LR and SPSA estimator, respectively. Both the estimators can benefit from the allocation and the Gaussain allocator has better performance than the Bernoulli allocator. The gradient of deep layer like layer 1 is more difficult to estimate than the shallow layer and the corresponding variance accounts for most of the variance of the whole model. Comparing Figures 6(a)-(c) to (g)-(i), estimation on the feed-forward network is easier than on the multi-head attention. In practice, it is recommended to add gradient clipping to the multi-head attention layer.

