# OpenReview forum: "FLOPS: Forward Learning with OPtimal Sampling"
_ICLR.cc/2025/Conference — ICLR 2025 Poster_

### Official Review · Reviewer_dE4A · 2024-10-20

**Soundness:** 2
**Presentation:** 1
**Contribution:** 3
**Rating:** 8
**Confidence:** 4

**Summary:**

This paper presents an approach for optimizing a differentiable sampler in forward learning in foundation models, supported by theoretical proofs and extensive empirical evaluations. The work lies at the intersection of zeroth-order optimization and sampling optimization.

**Strengths:**

1. The empirical evaluation is comprehensive, demonstrating significant performance improvements through query sampler optimization.

**Weaknesses:**

1. **Overall outline and structure**: The paper builds upon DeepZero and Mezo by devising an optimization allocation for forward learning. However, the idea of optimal allocation is not new in the ML context. Papers such as "Stochastic Optimization with Importance Sampling" or "A General Analysis of Example-Selection for Stochastic Gradient Descent" (and several derived works) have explored similar concepts. For me, the main difference here is the focus on forward learning (or zeroth-order optimization) rather than backpropagation. The authors are encouraged to: (1) Review and mention this existing research (strongly encouraged) and (2) Compare these methods with their proposed algorithm (encouraged). Addressing these points would strengthen the paper's contribution and contextualize it within the field.

2. **Introduction and framing**:
The decision to begin with biologically plausible algorithms (BioPA) seems unexpected and may not effectively frame the paper's contributions. Nevertheless, the general scheme could be unfolded more clearly:
     1. The citation of Jacot et al. for "learning high-level representation" appears unrelated in the BioPA context.
     2. Consider including more relevant BioPA works such as "Direct Random Target Projection" [1], "SoftHebb" [2], and "Counter-Current Learning" [3]

3. **Writing and proofreading**:
   - Correct typographical errors (e.g., "Current Literature" should be "Current literature")
   - Address factual inaccuracies (e.g., L48 states that the FF algorithm is only capable of training MLPs on MNIST, but results for CIFAR are also presented)
   - Provide explanations for abbreviations (e.g., SPSA, LR)

4. **Related Work section**: The first subsection could be restructured. When discussing backpropagation-free learning, it's typically in the context of multi-layered neural networks. Also, including evolution theory and particle swarm optimization seem tangential. I suggest reorganizing this section and incorporating the suggested papers for a more focused discussion.

References

[1] "Learning Without Feedback: Fixed Random Learning Signals Allow for Feedforward Training of Deep Neural Networks" (Frontiers in Neuroscience, 2021)

[2] "Hebbian Deep Learning Without Feedback" (ICLR 2023)

[3] "Counter-Current Learning: A Biologically Plausible Dual Network Approach for Deep Learning" (NeurIPS 2024)

**Questions:**

1. **Clarification on LR and OPS-LR**: The distinction between LR and OPS-LR is not clear. More explanation would be appreciated.

2. **Clarification on experiments**: Do the experiments include cross-validation with multiple random seeds? If not, please show the experiment results with multiple seeds with dataset cross-validation. If yes, please provide more details.

3. **Ablation studies**: The proposed algorithms update four parameters. What if only three or two of them are updated? Which parameters are dispensable for this process? Conducting these experiments would provide more insights into the paper's contributions.

---

> ### Comment · Reviewer_dE4A · 2024-11-24
> **Response to the authors**
>
> After reviewing the authors' responses and the revised manuscripts, I think the authors have addressed my concerns with additional supporting experiments. The revised version is now clear in definition and presentation, and thus I raise my score for this paper.

---

### Official Review · Reviewer_HWGc · 2024-10-25

**Soundness:** 3
**Presentation:** 3
**Contribution:** 3
**Rating:** 6
**Confidence:** 1

**Summary:**

This paper proposes an efficient query allocation strategy for forward-learning algorithms in gradient computation, reducing query usage by focusing on data points that need it most. Using a simplified objective and reparameterization, the authors introduce a lightweight query allocator that minimizes gradient estimation variance with low computational cost. Both experiments and theoretical analysis are provided.

**Strengths:**

1) The motivation is clear: optimizing the allocation of queries to effectively reduce computational overhead.
2) The experimental results show strong performance relative to the baselines.
3) The study provides both experimental and theoretical results, offering a well-rounded evaluation.

**Weaknesses:**

1) I am curious about why other methods that utilize all queries would perform worse than this method, that utilizes limited quries for each data.
2) The comparison of exact computational cost between equally using all queries for each data point and your allocation method is unknown. However, it is one of the main motivation.

Minor:
3) In the abstract, the phrase “propose to allocate the optimal number of queries over each data” isn’t entirely accurate, as a total query budget must be pre-defined rather than learning an optimal number. Perhaps rephrasing to “allocate the optimal number of queries within a set budget” would be clearer.
4) Table 2 is not well-formatted and appears misaligned.

**Questions:**

1) I am curious about why other methods that utilize all queries would perform worse than this method, that utilizes limited quries for each data.  Since MEZO is tailored to another baseline, using the same hyperparameters might not be entirely fair. Have you considered tuning MEZO with more available queries per data point for comparison?
2) What is the exact computational cost when using all queries for each data point compared to your allocation method under different budget constraints?

If reasonable answers are provided, I will consider rasing scores accordingly.

---

### Official Review · Reviewer_5nkq · 2024-10-29

**Soundness:** 3
**Presentation:** 2
**Contribution:** 2
**Rating:** 6
**Confidence:** 3

**Summary:**

The paper introduces FLOPS: Forward Learning with Optimal Sampling, which aims to improve the efficiency of gradient estimation in forward-only learning methods by optimally allocating computational resources (queries) across data points within each mini-batch. The approach is motivated by the limitations of backpropagation, particularly in settings where only forward passes are feasible or desirable, such as in black-box optimization scenarios.  With a simplified proxy objective and a reparameterization technique, the authors derive a novel plug-and-play query allocator with minimal parameters. Extensive experiments show the superior performance of this method. Theoretical analysis is also provided.

**Strengths:**

1. The idea of dynamically allocating different numbers of queries to each data point within a batch during training is novel, which is indeed a point that previous zeorth-order optimization (forward learning) methods have not considered.

2. The proposed method is intuitive. The approach of leveraging a Gaussian Allocator (GA) combined with a likelihood ratio method introduces a creative solution to minimize gradient estimation variance. Through appropriate approximations, the computational cost is effectively reduced. Theoretical analysis is also provided.

3. The experimental setup is extensive and reasonable, and the results are convincing. Both prompt tuning for large models and multimodal alignment for foundation models are promising application scenarios for zeroth-order (ZO) methods, and the proposed approach demonstrates good performance on these tasks.

**Weaknesses:**

1. Although the authors provide part of the source code, I believe the coding is not advisable. Specifically, the authors override nn.Linear to create a custom Linear class and similarly override nn.Conv2d to create a custom Conv2d class. This approach results in the proposed method being tied to a specific model architecture, making it difficult to adapt to other architectures. In fact, existing zeroth-order optimization methods, such as ZO-SGD [1], ZO-AdaMM [2], and DeepZero [3], all have core optimization logic that can be implemented by inheriting from torch.optim.Optimizer, thereby aligning with gradient-based methods like SGD. Alternatively, they can be integrated into a specific function for easier migration.

2. One important reason why zeroth-order optimization is suitable for large model prompt fine-tuning is that these methods do not require backpropagation, which significantly saves memory compared to gradient-based methods like SGD. However, in the experimental results presented in the main paper, only the fine-tuning results are provided, without comparing their memory usage with backpropagation and other zeroth-order optimization methods.

[1] Saeed Ghadimi and Guanghui Lan. Stochastic first-and zeroth-order methods for nonconvex stochastic programming. SIAM Journal on Optimization, 23(4):2341–2368, 2013.

[2] Xiangyi Chen, Sijia Liu, Kaidi Xu, Xingguo Li, Xue Lin, Mingyi Hong, and David Cox. ZO-AdaMM: Zeroth-order adaptive momentum method for black-box optimization. NeurIPS, 32, 2019.

[3] Chen A, Zhang Y, Jia J, et al. Deepzero: Scaling up zeroth-order optimization for deep model training[J]. arXiv preprint arXiv:2310.02025, 2023.

**Questions:**

1. Please refer to Weakness 1. Is it possible to integrate the proposed method into a callable class or function without rewriting model architectures like Linear and Conv2d to implement the specific add_noise operation?

2. Please refer to Weakness 2. Could you provide detailed memory usage for different methods during training for a more thorough comparison? I noticed that the code uses a repeat operation to expand the batch for varying numbers of queries on different data. Does this operation increase memory usage?

3. The authors mention in the main text that 'All the methods in the experiments use the same query budgets, except for Mezo, which uses only 2 queries per data point in accordance with its original memory-efficient setting.' However, could you provide a more detailed comparison of runtime (e.g., clock time) compared to other ZO methods and the BP baseline?

---

### Official Review · Reviewer_oVeC · 2024-11-03

**Soundness:** 2
**Presentation:** 1
**Contribution:** 2
**Rating:** 3
**Confidence:** 3

**Summary:**

This paper examines perturbation-based gradient computation methods tailored for forward-only learning. The authors introduce an optimal sampling strategy based on a Gaussian Allocator designed to maximize performance improvements incrementally. They evaluate this approach using pretrained transformers and demonstrate that it outperforms selected baseline methods.

**Strengths:**

1. The empirical accuracy results for ViT and CLIP appear to surpass those of the baselines. The authors have also conducted essential ablation studies to further validate their findings.

2. While the text in Figures 2 and 3 is smaller than the standard text size, making it challenging to read, the color combinations used in these figures are visually appealing.

**Weaknesses:**

1. Sections 3 and 4 introduce several undefined annotations, leading to ambiguity in the exposition. For instance, Line 148 mentions a term "a" whose role and relation to Equation (1) are unclear—is it a distribution, a hidden representation, or something else? Furthermore, "G(·)" on Line 160 and "y_j" on Line 164 are undefined, with no clarification of the indexing or distribution from which j is sampled. Additionally, Equation (9)'s term "K" lacks a defined scope. The abbreviation "LR" is also used without definition. The paper introduces many terms that, while relevant, obscure the main contributions. I recommend a thorough review of these sections to clarify the foundational concepts and distinctly outline the problems the paper aims to solve.

2. The rationale behind formulating the optimization of the query allocator as maximizing Equation (5) is not sufficiently clear. A deeper analysis of this formulation is necessary. Moreover, the paper mentions only the initialization of a Gaussian Allocator, which might suggest a broader applicability than is actually the case. Either a comparative analysis of different allocators should be included, or the focus on the Gaussian allocator should be explicitly stated in the abstract and introduction to manage expectations.

3. I have several concerns regarding the experimental results presented in Table 1, which utilizes a pretrained ViT network. Firstly, the rationale for the notably small number of queries for baseline methods, such as only 2 for Mezo, is not explained. What is the expected number of queries for your OPS-LR model in comparison? In Line 344, you mention alignment with Mezo's original memory-efficient setting, but detailed statistical data on memory consumption and training time for both baseline methods and your approach are lacking. Additionally, the experiments are conducted on pretrained models like ViT and CLIP, but the performance of models trained from scratch is not shown. It is crucial to demonstrate whether OPS-LR offers any advantages over these baselines when applied to models not pretrained. Furthermore, the validation datasets used are generally small, with all except ImageNet containing under 150,000 images. There is also concern that using pretrained transformers on the well-known ImageNet dataset could lead to performance biases due to data leakage.

**Questions:**

For enhanced clarity, please focus on the weaknesses section, which includes my raised questions.

Concerning reproducibility, I reviewed the supplementary material, presumably the code. However, the absence of a README file and the presence of numerous extraneous files make it challenging to determine which files are essential. Additionally, the provided code lacks meaningful comments and contains a lot of debug information, which further complicates understanding its logic. If the authors aim to demonstrate reproducibility through the attached code, I recommend including a clean version of the code with detailed instructions in a README file to at least guide reviewers through the main logic.

---

### Meta-Review · Area_Chair_6mt4 · 2024-12-16

**Metareview:**

**Summary:** The paper proposes a method for perturbation-based forward gradient learning. The authors claim that the technique has lower variance and is more scalable than the previous approaches. Their method relies on importance estimation for data points and unifies the previous approaches. They provide theoretical results for optimality and show results on foundation models.

**Strengths:** The reviewers believe that the idea is novel and the approach is intuitive. The method combines Gaussian Allocator (GA) with a likelihood ratio for importance estimation. The experimental results are thorough and promising and might be the strongest aspect of the paper. Some results such as ViT and CLIP appear to surpass those of the baselines, and the additional results on foundational models show good performance on these tasks. The reviewers do not raise any concerns about the theoretical results.

**Weaknesses:** Some of the notations and definitions are ambiguous. The motivations are unclear at parts, for instance, the query allocator as maximizing Equation (5). Memory usage compared to backpropagation and other zeroth-order optimization methods is not provided. Also, Reviewer HWGc claims that ``the ​​exact computational cost between equally using all queries for each data point and your allocation method is unknown''.

**Decision:** The paper addresses an important and practical problem in ML, which has immediate applications for more efficiently training large-scale models. The authors have done an excellent job of providing experimental evidence for the applicability of their method. The main weaknesses are related to 1) writing, proofreading, and presentation, which the authors have improved greatly during the rebuttal period, 2) some missing experiments and justifications, which the authors provide further additional results to address. Thus, based on the vote by the majority of the reviewers, and the authors’ response during the rebuttal period, I recommend acceptance.

**Additional Comments On Reviewer Discussion:**

As mentioned before, the authors did an excellent job providing further evidence and clarification to the points raised by the reviewers. Reviewer oVeC has the lowest score but unfortunately did not participate in the rebuttal period. As far as I can tell, the authors mostly addressed their concerns.

---

### Decision · Program_Chairs · 2025-01-22

Accept (Poster)